# Protocol for surgical and non-surgical treatment for metacarpal shaft fractures in adults: an observational feasibility study

Rowa Taha ,[1] Paul Leighton,[2] Chris Bainbridge,[3] Alan Montgomery,[4] Tim Davis,[5] Alexia Karantana[6]

¹Academic Orthopaedics, Trauma & Sports Medicine, University of Nottingham School of Medical and Surgical Sciences, Nottingham, UK
²Division of Primary Care, University of Nottingham, Nottingham, UK
³Pulvertaft Hand Centre, Royal Derby Hospital, Derby, UK
⁴Nottingham Clinical Trials Unit, University of Nottingham, Nottingham, UK
⁵Trauma and Orthopaedics, Nottingham University Hospitals NHS Trust, Nottingham, UK
⁶Surgery, University of Nottingham Faculty of Medicine and Health Sciences, Nottingham, UK

**Correspondence to**
Rowa Taha;
rowa.taha@nottingham.ac.uk

## ABSTRACT

**Introduction** Metacarpal shaft fractures (MSF) are common traumatic hand injuries that usually affect young people of working age. They place a significant burden on healthcare resources and society; however, there is a lack of evidence to guide their treatment. Identifying the most beneficial and cost-efficient treatment will ensure optimisation of care and provide economic value for the National Health Service. The aim of this study is to assess the feasibility of a randomised controlled trial comparing surgical and non-surgical treatment for MSF in adults.

**Methods and analysis** This is a multicentre prospective cohort study, with a nested qualitative study consisting of patient interviews and focus groups, and an embedded factorial randomised substudy evaluating the use of text messages to maximise data collection and participant retention. The outcomes of interest include eligibility, recruitment and retention rates, completion of follow-up, evaluation of primary outcome measures, calculation of the minimal clinically important difference (MCID) for selected outcome measures and establishing the feasibility of data collection methods and appropriate time-points for use in a future trial. Data will be captured using a secure online data management system. Data analyses will be descriptive and a thematic inductive analysis will be used for qualitative data. Minimum clinically important effects for each patient-reported outcome measure will be estimated using anchor-based responsiveness statistics and distribution-based methods.

**Ethics and dissemination** This study has received ethical approval from the Research Ethics Committee and the Health Research Authority (REC reference 20/EE/0124). Results will be made available to patients, clinicians, researchers and the funder via peer-reviewed publications and conference presentations. Social media platforms, local media and feedback from the Patient Advisory Group will be used to maximise circulation of findings to patients and the public.

**Trial registration number** ISRCTN13922779.

## Strengths and limitations of this study

► Provides comprehensive information to inform the design and implementation of a future trial comparing treatments for metacarpal shaft fractures (MSF), including evaluation of multiple outcome measures and calculation of minimal clinically important difference to inform future sample size calculations.
► Provides complementary person-centred insight into MSF and research design, conduct and delivery through embedded qualitative assessments.
► Evaluates effective text message strategies for maximising data collection and retention in research studies.
► A detailed cost evaluation is a particular strength of the study and will deliver a comprehensive analysis of the costs of treatments available for MSF.
► Limited assessment of randomisation.

decade of life,[1 5] with the fourth and fifth metacarpal most commonly injured.[1–6]

In 2016, there were 23.5 million accident and emergency (A&E) department attendances in the UK with fractures being the second most common reason for presentation.[7] As hand fractures make up 25% of all A&E attendances[8] and MSF comprise 18%–31% of hand fractures,[1–4] MSF therefore place a significant burden on healthcare resources.

MSF predominantly affect those of working age[2–4 8 9] and are thus associated with significant cumulative morbidity.[10] Missed time off work significantly increases the economic burden of MSF.[10] However, there is no UK-specific data on the healthcare-associated costs, socioeconomic, or societal costs of these injuries.

There is wide variability in the management of MSF. MSF can be managed non-surgically with appropriate reduction and immobilisation,[11 12] or surgically using a variety of

## INTRODUCTION

Metacarpal shaft fractures (MSF) are common traumatic hand injuries reported to represent 18%–31% of hand fractures.[1–4] They usually affect young adult males, often in the third

different techniques, including Kirschner-wires (K-wires), intraosseous wires, interfragmentary compression screws, plates or external fixators.[13]

A systematic review undertaken in 2019 of treatment interventions for MSF identified 699 records and no randomised controlled trials (RCTs) comparing surgical to non-surgical treatments.[14] The only retrospective cohort study had several key limitations including small patient numbers, low follow-up rate (17%) and lack of use of a patient-reported outcome measure (PROM) validated in MSF.[15] A search of the WHO ISCTRP portal also revealed no ongoing or registered trials worldwide.[14]

### Rationale for study

There is a lack of good quality, large comparative trials to guide the treatment of MSF. There are no published or ongoing RCTs or cohort studies comparing surgical versus non-surgical treatment for MSF. In addition, though the use of PROMs in both the clinical and research setting has increased in recent years, there is no evidence of reliability, validity and responsiveness of PROMs in MSF.

There are several gaps in the literature:

▶ No consensus on acceptable parameters of deformity or displacement, leading to widespread variation in treatment
▶ No core outcome sets for hand trauma, so we do not know which outcome measures are best suited for the study of MSF
▶ No qualitative data exploring patient experience of MSF and their treatment
▶ No evaluation of the cost-effectiveness of treatment modalities for MSF
▶ No high-quality published evidence comparing treatment modalities for MSF

The lack of existing evidence supports the need for a well-designed, pragmatic, multicentre RCT to identify the most beneficial and cost-efficient treatment for MSF in adults. This study aims to assess the feasibility, acceptability and practicality of such a trial by providing information about study design, number of eligible patients, recruitment, completion of follow-up, selection of appropriate outcome measures, assessment of minimal clinically important difference (MCID) for selected outcome measures, costs of treatments and measures to optimise recruitment, engagement and retention in a future trial.

### Study objectives and purpose

The overall purpose of this study is:

1. To investigate the feasibility and acceptability of conducting a multicentre RCT to assess the clinical and cost-effectiveness of surgical and non-surgical treatment for MSF in adults.
2. To provide complementary, detailed and person-centred insight that will inform RCT design through the identification of barriers to participation among patients with MSF, and to develop novel solutions to engage these cohorts in research.

The objectives of the study (table 1) were developed at a MSF Consensus Workshop, held by the Centre for Evidence Based Hand Surgery in Nottingham, November 2018 involving patients, clinicians, therapists and clinical trials methodologists.[16]

## METHODS AND ANALYSIS
### Study configuration

FACTS is an observational, multicentre, prospective cohort study. Treatments will not be randomly allocated; patient care will be determined in the usual way as per the treating clinician.

Patients who meet the eligibility criteria (table 2) will be recruited from hand fracture clinics at participating National Health Service (NHS) hospitals. Where participants have multiple MSF with potentially variable fracture patterns, a single digit will be selected as the 'study finger'. Participants with additional hand or upper limb injuries will be included and the latter recorded. Written informed consent will be obtained from all participants by an appropriately trained research associate prior to entering the study. Thereafter, participants will be reviewed in a face-to-face clinic visit at 6 weeks and remotely at 3 and 6 months. This was supported by previous studies of MSF reporting full range of motion in almost all patients by 6 months.[17] Furthermore, previous studies in similar patient cohorts with follow-up of 12 months or longer suffered high drop-rates or struggled to recruit adequate numbers of participants. A participant pathway flowchart is illustrated in figure 1.

A nested qualitative study consisting of two elements, patient interviews and focus groups, will be conducted to provide patient-centred insight into study procedures and explore the individual impact of the injury. Participants will be selected from the prospective cohort study, using purposive sampling that prioritises young males, and further written informed consent separately sought for this element of the study.

A detailed cost evaluation to establish the costs of treatments for MSF through representative micro-costing will be undertaken. Resource use directly linked to the MSF and its sequela and/or complications over the 6 months of follow-up will be recorded for each participant. Unit cost data will be obtained from national databases such as NHS reference costs, the British National Formulary (BNF) and Personal Social Services Research Unit (PSSRU) Costs of Health and Social Care.[18]

A two by two by two factorial randomised substudy will be nested within the main cohort study. Once participants have consented to the cohort study or qualitative study, they will be randomised to the following interventions to evaluate the use of text messages in maximising data collection and participant retention; frequency of text messages—either fortnightly or monthly; two-way communication—text message requiring a response from the participant versus a notification message only; and personalisation—a personalised message versus a standard automated message.

**Table 1** Study objectives

| How objective will inform the definitive trial | Objective |
|---|---|
| Recruitment for a future trial | 1. Define eligibility criteria for the future trial, which correctly identify appropriate patients for whom a treatment decision is suitable<br>2. Estimate the proportion of referred NHS patients who meet these eligibility criteria<br>3. Assess recruitment and retention rates |
| Outcomes for use in a future trial | 4. Evaluate outcomes for use as primary and secondary outcomes<br>5. Calculate minimal clinically important difference for the proposed primary outcome measure<br>6. Investigate feasibility of collecting outcome data frequently, in order to capture subtle improvements in patient-assessed or clinician-assessed outcomes |
| Follow-up | 7. Estimate follow-up and outcome completion rates for both clinic and remotely assessed outcomes<br>8. Explore optimum time-points for follow-up |
| Sample size calculation | 9. Estimate the sample size required for a definitive study |
| Use of remote assessments in a future trial | 10. Evaluate the utility and acceptability of remote completion of health resource use questionnaires to assess the impact of care on health service use and productivity<br>11. Evaluate the utility and acceptability of remote, electronically administered patient assessments |
| Economic assessment | 12. Inform the design of a future cost-effectiveness analysis by exploring the costs of treatment modalities through capture of NHS resource use and representative micro-costing |
| Patient-centred insight into research design, conduct and delivery | 13. To explore participant experience of MSF, treatment and recovery<br>14. To explore participant experience of research processes and study burden associated with outcome measures<br>15. To gain recommendations on future study design and mechanisms to facilitate study delivery |
| Facilitate engagement and retention among patients with MSF in a future trial | 16. To explore the use of health technology applications and social media in optimising participation and engagement in research |

MSF, metacarpal shaft fractures; NHS, National Health Service.

## Treatment groups

Participants will enter a group according to their primary mode of treatment on enrolment into the study. Patients will undergo standard care as per their treating clinician. This will be decided by their clinical care team and will not be affected by their involvement in the study. Treatment groups will be defined as follows;

Surgery: any surgical treatment, defined as insertion of metal via an open or percutaneous approach in an operating theatre, such as open reduction internal fixation, closed reduction internal fixation, intramedullary fixation or wiring, extramedullary wiring or external fixation.

Non-surgical treatment: any 'non-surgical' treatment, defined as regimens with or without reduction (partial or complete) of the fracture, any type of splinting or cast, and/or immediate or delayed mobilisation delivered in a clinic or therapy room environment.

## Outcomes

Participants will be invited to attend a research clinic at 6 weeks following injury. During this visit, clinical assessments of their hand including range of movement, grip strength and presence of rotational deformity or extensor lag will be recorded. Radiographs taken as part of routine clinical care will also be reviewed. The location, fracture morphology, amount of shortening, angulation and presence of step-off deformity on initial radiographs at presentation will be recorded. Complications will be identified by review of the patients' healthcare records at 6 weeks, 3 months and 6 months. Cosmesis will be assessed from question 10 of the Patient Evaluation Measure (PEM), at 6 weeks, 3 months and 6 months. Participant-reported complications will be recorded at the 6-week research clinic visit.

The following PROMs were selected for use following discussions between clinicians, patients, therapists and researchers at the MSF consensus workshop.[16] They were prioritised due to their ease of use and validity and will be collected at baseline, 6 weeks, 3 months and 6 months.

### Patient Evaluation Measure

The PEM consists of 11 items relating to hand function and appearance, each scored from one to seven from best/normal to worst. It is the PROM of choice in the British Society for Surgery of the Hand (BSSH) National UK Hand Registry (www.ukhr.net) and has been demonstrated to be reliable, valid and responsive for assessing hand disorders.[19 20]

| Table 2 | Eligibility criteria |
|---|---|
| Inclusion criteria | ▶ Adults 16 years or older<br>▶ Radiologically confirmed metacarpal shaft fracture<br>▶ Acute metacarpal shaft fracture affecting the index to little finger(s), presenting within 10 days of injury<br>▶ Willing and able to give informed consent<br>▶ Ability to understand English |
| Exclusion criteria | ▶ Fracture(s) of the thumb<br>▶ Fractures extending into the joint surface<br>▶ Fracture(s) of the metaphyseal base and/or neck of the metacarpal<br>▶ Fracture(s) associated with dislocation at the carpometacarpal joint or other adjacent joint dislocation<br>▶ Open fractures<br>▶ Undisplaced fractures, defined as those with a visible fracture line on radiographs but anatomical alignment, that is, the bone fragments remain aligned with no evidence of movement of the fracture fragments on anteroposterior, lateral or oblique radiographs<br>▶ Patients who would not be able to adhere to study procedures or complete the study questionnaires |

## Patient-reported outcomes measurement information system upper extremity

The PROMIS UE (PROMIS UE Item Bank V.2.0) is a computerised adaptive test, developed by the United States National Institute of Health using item response theory. It has been validated in upper limb fracture and is designed to minimise patient burden and theorised to measure latent traits more precisely than existing PROMs.[21] For participants who do not engage with electronic means, a paper-based short-form alternative version is available.

## Shortened Disabilities of the Arm, Shoulder and Hand Outcome Measure (QuickDASH)

The Disabilities of the Arm, Shoulder and Hand Score (DASH) is the most commonly used PROM in hand and wrist trauma and has consistently demonstrated good reliability, validity and responsiveness in several psychometric studies.[22] It consists of 11 items, developed from the original 30-item DASH to improve practicality and eliminate item redundancy.[23]

## European Quality of Life Questionnaire (EQ-5D-5L)

The EQ-5D-5L is a validated, generalised and standardised instrument comprising a Visual Analogue Scale (VAS) measuring self-rated health and a health status instrument, consisting of a five-level response for five domains related to daily activities.[24] This standardised measure of health status provides a simple, generic measure of health for clinical and economic appraisal.[25]

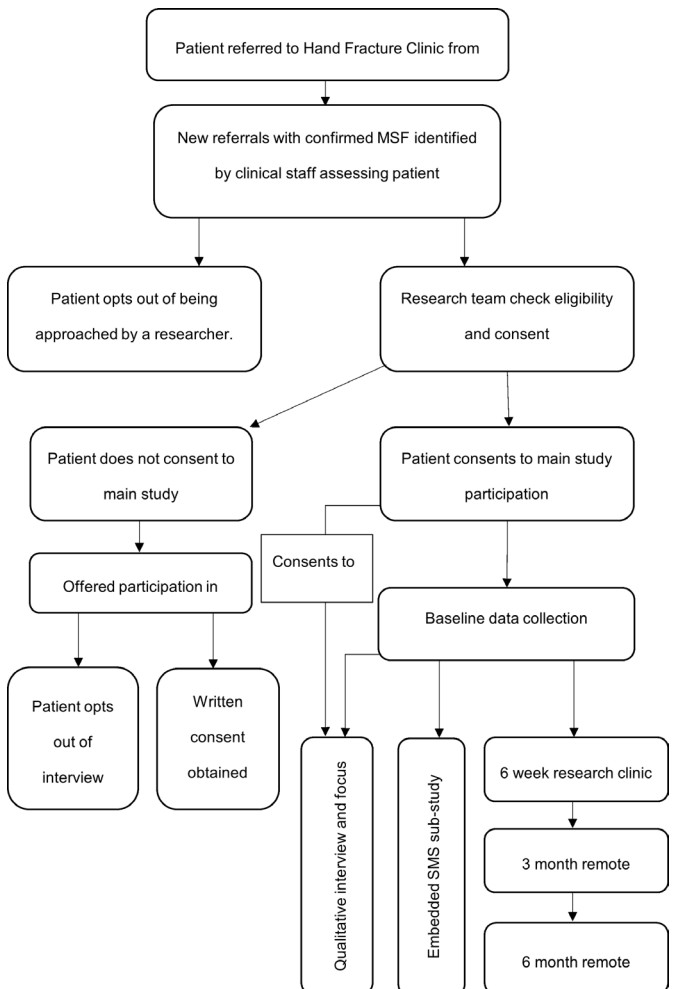

**Figure 1** Study flow diagram. MSF, metacarpal shaft fractures. SMS, Short Message Service.

## Global Rating of change (GROC)

The GROC scale is commonly used as an anchor when calculating MCID. It is designed to quantify a patient's improvement or deterioration over time, thus providing a means of measuring self-perceived change in health status.[26] A 7-point scale ranging from −3 (very much worse) to +3 (very much better), with 0 indicating 'unchanged', will be used.

## Sample size and Justification

As this is a feasibility study, formal sample size calculations for between-group comparisons are not appropriate. However, we will seek to include as many as 84 participants in the study, aiming for comparable numbers in each treatment group. This sample size will enable estimation of recruitment fraction with margin of error (half width of 95% CI) of <9 percentage points and of proportions estimated from the recruited sample, such as completeness of follow-up, to within 13 percentage points.

A purposive sample of 12–16 participants, who indicate a willingness to be interviewed and/or attend focus groups, will be recruited to the qualitative study. Participants who decline to participate in the main cohort study will also be invited for interview.

## Analysis of outcomes

The outcomes of interest include feasibility outcomes relating to; assessment of eligibility, recruitment and retention rates; completion of follow-up; evaluation of outcome measures and calculation of the MCID for selected outcome measures using quantitative and qualitative assessments; and establishing the feasibility of data collection methods and appropriate time-points for use in a future trial. To address these feasibility aims, data analyses will primarily be descriptive with 95% CIs to quantify uncertainty in estimates where appropriate.

## MCID evaluation

Minimum clinically important differences for each PROM at 3 months will be estimated using three anchor-based responsiveness statistics: (1) standardised response mean; (2) effect size and (3) Guyatt's Responsiveness Index. An estimation of the MCID will also be calculated by distribution-based methods, using the SE of measurement, SD and effect size. The minimal detectable change will also be calculated to ensure the MCID is greater than the measurement error of the PROM. The sensitivity and specificity of the PROM will be assessed in conjunction with the MCID, as calculated above.

## Costing of interventions

Direct observation of procedures will be used to produce a 'micro-cost' estimate for surgical and non-surgical treatments by combining resource use with unit costs provided by hospital finance departments. Duration of each procedure, theatre staffing, consumables, imaging, supplementary devices, postoperative recovery time and rehabilitation inputs will be recorded from primary sources, such as theatre log systems and patients' electronic and paper clinical records. Standard unit costs will be used to estimate NHS costs of care in the 6-month post-treatment. Unit cost data will be obtained from national databases such as the BNF and PSSRU Costs of Health and Social Care 2019.[18]

There is a need to define the economic morbidity of MSF in terms of the costs of treatment and associated resource use, productivity losses and additional costs incurred by patients during the course of their treatment and recovery. Identifying the costs of treatments will help to establish the feasibility of collecting utility and cost data in a future trial, as well as devising methods for accurately collecting such data to ensure robust cost-effectiveness comparisons in a future trial.

## Qualitative analysis

A thematic, inductive approach as described by Braun and Clarke[27] will be used to analyse interview and focus group data. We will adopt a six-phase systematic approach consisting of (1) data familiarisation—reading and re-reading the data; (2) generation of initial codes—generating succinct labels (codes) that identify important features of the data that might be relevant to answering the research question; (3) identification of themes through merging and grouping of codes—to identify significant broader patterns of meaning (potential themes) and collating data relevant to each theme; (4) review of generated themes—checking themes against the dataset and refining them where necessary; (5) defining and naming themes—developing a detailed analysis of each and deciding the informative name for each theme and (6) finalisation of themes and generation of a final report—weaving together the analytic narrative and data extracts, and contextualising the analysis in relation to existing literature. NVivo 12 or above Pro software (QSR International Pty, Victoria, Australia) will be used for analysis.

## Patient and public involvement statement

Patients and the public have played a central role in selecting the management of MSF as a key research priority and developing the proposed study. The Hand Surgery Research Prioritisation workshop, attended by clinicians, therapists and patients, recommended the management of MSF as a key research priority.[28] This was ratified by the James Lind Alliance Priority Setting Partnership on Common Conditions Affecting the Hand and Wrist.[16] This joint collaboration with the BSSH involved 261 individuals, of which 41% were patients/carers and 59% were clinicians.[16]

Furthermore, a MSF Consensus Workshop was held in Nottingham in 2018, bringing together patients with MSF, clinicians, therapists and researchers to share their experiences and develop the PICO framework for a future multicentre trial.[16] Group discussions informed the eligibility criteria and identified areas of focus for the feasibility work. Selection of patient-centred outcome measures, timing of assessments and follow-up for the proposed study were discussed. A variety of outcome measures were reviewed and the QuickDASH and PROMIS-UE were subsequently added. One clinic visit in addition to virtual follow-up was included and follow-up was adjusted to 3 and 6 months following patient discussions. Feedback from patients who attended the workshop has informed all aspects of the research including research design, choice of outcome measures and length and location of follow-up.

The study protocol, participant information sheets and consent forms have been reviewed by PPI members, with feedback provided to optimise their utility. We have set up a Metacarpal Shaft Fracture Patient Advisory Group to inform the design, delivery and output of the research. An electronic PPI platform has been created on the Centre for Evidence Based Hand Surgery (CEBHS) website to encourage regular input from patients/public.

Feedback from the Metacarpal Shaft Fracture Patient Advisory Group will guide distribution of findings to the public. This will include, but not be limited to, newsletters, local media outlets, events and plain English summaries of all published journal articles.

Incorporating patient and public involvement in all aspects of the research pathway, from research design to co-development and review of all study documents, supports our commitment to ensuring sustained and meaningful PPI throughout the study.

### Study management
The study is funded by a National Institute for Health Research Doctoral Fellowship awarded to Miss Rowa Taha (NIHR300197). It is sponsored by the University of Nottingham and will be managed and co-ordinated from the Centre for Evidence Based Hand Surgery (CEBHS), University of Nottingham.

## ETHICS AND DISSEMINATION
### Ethics committee and regulatory approvals
This study has received approval from the Research Ethics Committee (REC), the respective NHS Research & Development (R&D) departments and the Health Research Authority (HRA) (REC reference 20/EE/0124). It will be conducted in accordance with the ethical principles that have their origin in the Declaration of Helsinki[29]; the principles of Good Clinical Practice[30] and the UK Department of Health Policy Framework for Health and Social Care, 2017.[31]

### Informed consent
The process for obtaining participant informed consent will be in accordance with the REC guidance and Good Clinical Practice.[30] The investigator or their nominee and the participant or other legally authorised representative shall both sign and date the informed consent form before the person can participate in the study. Written informed consent will be separately sought for taking part in the qualitative study.

### Safety considerations and data protection
There are no significant safety issues with this study. The study is observational and does not interfere with the routine clinical care pathway. The treatments participants receive are widely available within the National Health Service and will not be altered by taking part in the study. The questionnaires, interviews and focus groups are neither burdensome nor contain sensitive questions and all reasonable expenses, such as travel and parking costs, associated with attending research visits will be fully reimbursed.

All study staff and investigators will endeavour to protect the rights of the study's participants to privacy and informed consent and will adhere to the Data Protection Act, 2018.[32] Case report forms will be held securely and access to the information will be limited to the study staff, investigators and relevant regulatory authorities. Computer held data including the study database will be held securely and password protected. All data will be stored on a secure dedicated web server. Access will be restricted by user identifiers and passwords (encrypted using a one-way encryption method).

### Dissemination
Results of this study will be reported fully and made publicly available when the research has been completed. The outcomes of the study will be published in suitable peer-reviewed journals and will be reported locally, nationally and internationally in the form of presentations. Reporting will be in compliance with Strengthening the Reporting of Observational Studies in Epidemiology (STROBE) guidelines.[33] In order to fulfil reporting guidelines, a copy of the research paper will also be sent to the National Institute for Health Research (NIHR) programme issuing the funding contract.

The findings will be presented at national and international meetings of relevant scientific societies. We will also publish key findings on the CEBHS website, and via the 'Hand Evidence Updates', distributed by the CEBHS to over 800 national and international members.

Social media platforms will also be used to maximise dissemination of key findings as supported by evidence from the Nottingham Clinical Trials Unit.[34 35]

**Contributors** The study concept and design was conceived by RT, PL, AM, TD and AK. PL advised on qualitative study design, data collection and analysis considerations. CB advised on study design, clinical assessments and data collection. RT prepared the first draft of the manuscript. All authors provided edits and critiqued the manuscript for intellectual content.

**Funding** This work was supported by the National Institute for Health (NIHR) research grant number NIHR300197 and the British Society for Surgery of the Hand (BSSH).

**Disclaimer** The views expressed in this publication are those of the author(s) and not necessarily those of the NHS, the National Institute for Health Research or the Department of Health and Social Care.

**Competing interests** RT, AK and AM report grants from the National Institute for Health Research (NIHR) and the British Society for Surgery of the Hand (BSSH). This protocol is independent research supported by the National Institute for Health Research (NIHR Doctoral Fellowship - Stage 2, Miss Rowa Taha, NIHR300197).

**Patient and public involvement** Patients and/or the public were involved in the design, or conduct, or reporting, or dissemination plans of this research. Refer to the Methods section for further details.

**Patient consent for publication** Not required.

**Provenance and peer review** Not commissioned; externally peer reviewed.

**ORCID iD**
Rowa Taha http://orcid.org/0000-0002-2403-5107

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
