## [Reviewer comments · BMJ Open]

ARTICLE DETAILS

TITLE (PROVISIONAL)	Protocol for surgical and non-surgical treatment for metacarpal shaft fractures in adults: an observational feasibility study
AUTHORS	Taha, Rowa; Leighton, Paul; Bainbridge, Chris; Montgomery, Alan; Davis, Tim; Karantana, Alexia

VERSION 1 – REVIEW

REVIEWER	Belloti, João Federal University of São Paulo (UNIFESP/EPM), Orthopedics and Traumatology - Division of Hand Surgery and Upper Limb
REVIEW RETURNED	20-Jan-2021

GENERAL COMMENTS	This cohort study protocol aims to evaluate the functional outcome of patients with fractures of the metacarpal diaphysis (2-5) treated with surgical or non-surgical treatment. The clinical question of the protocol is relevant due to the absence of conclusive evidence on various aspects of the treatment of these fractures. The methodological design is adequate, however, some points must be revised: - In the eligibility criteria, it is not clear whether patients with multiple diaphyseal fractures (2 or more pasterns) will participate similarly it is not clear whether patients with associated hand or wrist fractures will be included.- Regarding the assessment of outcomes, it was not clear whether complications in the surgical and non-surgical groups will be assessed.- There is no description of how patients with complications who need reintervention or change in the form of treatment will be allocated (eg fractures with unacceptable residual rotational deformity that require surgery)
--

REVIEWER	Tetsworth, Kevin Royal Brisbane and Women's Hospital
REVIEW RETURNED	14-Feb-2021

GENERAL COMMENTS	Thank you for inviting me to review this study protocol, titled the "Protocol for surgical and non-surgical treatment for metacarpal shaft fractures in adults: an observational feasibility study". You (collectively, the authors) and your collaborators have obviously put in a tremendous amount of time and energy to carefully consider the many aspects necessary to address the question at hand. Needless to say, the proposed study protocol is remarkably mature, and the authors are to be congratulated for the care they have taken to design the study. Although in my opinion
--

	there are more pressing issues of greater interest and concern, having selected an area of your own interest you have certainly taken on the project with gusto and the proposed protocol is methodologically robust; the likelihood of a successful outcome is high. After careful review, I have few if any recommendations to improve the protocol, only the following comments that the authors are invited to address: (1) Definitive specialist statistical review would be prudent to confirm my opinion that the techniques are in fact appropriate. (2) I would suggest extending the timeline to a full year follow up, and am puzzled why the 6 month review was agreed upon. If you could be kind enough to add a sentence or two in the Methods to justify this choice it would perhaps clarify your decision. Normally most orthopaedic trauma outcomes require a 12 month review to allow adequate time to recover fully, and to have a reasonable expectation that they have achieved the maximal benefit from the therapeutic intervention employed. (3) Page 11, lines 53-54: the study should already be overloaded with young male; in my opinion you would need justification to further emphasise this specific group. (4) Page 17, line 30: I question the wisdom of having civilians/non-medical personnel so actively involved in the study design. Could you please provide a statement to again justify why you have changed what I would consider an internationally recognised and accepted standard (minimum 12 months review) as to what would be an adequate follow up period.
--	--

VERSION 1 – AUTHOR RESPONSE

Reviewer 1 comment: In the eligibility criteria, it is not clear whether patients with multiple diaphyseal fractures (2 or more pasterns) will participate similarly it is not clear whether patients with associated hand or wrist fractures will be included.

Author's response: We have added a clarification regarding the inclusion of participants with multiple metacarpal shaft fractures in paragraph 2 of the methods and analysis. We have added a sentence clarifying that patients with additional hand or upper limb injuries are not excluded from the study (page 10, line numbers 46 – 53).

Review 1 comment: Regarding the assessment of outcomes, it was not clear whether complications in the surgical and non-surgical groups will be assessed.

Author's response: We appreciate that complications are an important outcome of interest in surgical studies and have included details of how they will be assessed in the outcomes section, methods and analysis (page 13, line numbers 30 - 36).

Reviewer 1 comment: There is no description of how patients with complications who need reintervention or change in the form of treatment will be allocated (eg fractures with unacceptable residual rotational deformity that require surgery).

Author's response: As this is an observational study, there will be no allocation of treatment. As stated on page 12, lines 47 – 53, participants' care will be decided in the usual way by the treating clinician. Participants will enter a group according to their primary mode of treatment as decided by their clinical care team on enrolment into the study. This will not be affected by their involvement in the study. Any change in treatment, for example patients who are initially treated non-surgically and go on to require

surgical treatment, will be recorded.

Reviewer 2 comment: Definitive specialist statistical review would be prudent to confirm my opinion that the techniques are in fact appropriate.

Author's response: The statistical aspects of this study have been developed and reviewed by a statistician with expertise in the application of statistical methods to the design, conduct and analysis of pragmatic, multi-centre late phase randomised trials; Prof. A. Montgomery, Professor of Medical Statistics and Clinical Trials, University of Nottingham.

Reviewer 2 comment: I would suggest extending the timeline to a full year follow up, and am puzzled why the 6 month review was agreed upon. If you could be kind enough to add a sentence or two in the Methods to justify this choice it would perhaps clarify your decision. Normally most orthopaedic trauma outcomes require a 12 month review to allow adequate time to recover fully, and to have a reasonable expectation that they have achieved the maximal benefit from the therapeutic intervention employed.

Author's response: Six months was selected as the study endpoint in this feasibility study as, unlike many other lower limb or upper limb fractures, the majority of patients following uncomplicated metacarpal shaft fractures are expected to have regained full function by six months post-injury. This is supported by findings by Macdonald et al. 2014, who demonstrated a full range of motion in almost all patients with spiral metacarpal shaft fractures by six months, and all had returned to work within two to eight weeks of injury. Longer follow-up in this patient population must be balanced against participant burden, participants' cost and resources, as well as the significant risk of drop-out. As this is a feasibility study, should we find that the hypothesis regarding rate of recovery is not supported by the outcome data, the design of the future trial will be informed accordingly.

Reviewer 2 comment: Page 11, lines 53-54: the study should already be overloaded with young male; in my opinion you would need justification to further emphasise this specific group.

Author's response: This method of purposive sampling is frequently used in qualitative research and will ensure that the interview and focus group sample broadly reflects general patterns in the epidemiology of metacarpal shaft fractures, as they predominantly affect younger males.

Reviewer 2 comment: Page 17, line 30: I question the wisdom of having civilians/non-medical personnel so actively involved in the study design. Could you please provide a statement to again justify why you have changed what I would consider an internationally recognised and accepted standard (minimum 12 months review) as to what would be an adequate follow up period.

Author's response: National funding bodies in the United Kingdom specify the involvement of patient and public involvement (PPI) in the co-design, development and implementation of clinical research. This study complies with the UK Standards for Public Involvement in research, as specified by the funder, the National Institute for Health Research (NIHR). Research supported by the NIHR must comply with UK Standards for Public Involvement in research, to help researchers and organisations improve the quality and consistency of public involvement in health and care research.

We thank you for your kind consideration of our paper.